# A Bibliometric Analysis of Research into Internet Gaming Disorders in Korea

**DOI:** 10.3390/ijerph20053786

**Published:** 2023-02-21

**Authors:** Melvyn W. B. Zhang, Seon Young Park

**Affiliations:** 1Lee Kong Chian School of Medicine, Nanyang Technological University, Singapore 308232, Singapore; 2National Addictions Management Service, Institute of Mental Health, Singapore 539747, Singapore; 3Department of Neuropsychiatry, Seoul National University Hospital, Seoul 03080, Republic of Korea; 4Department of Human Systems Science, Seoul National University, Seoul 03080, Republic of Korea

**Keywords:** internet gaming disorder, video gaming, behavioural addictions, South Korea, adolescents, bibliometric review

## Abstract

After the concept of “Internet addiction” was first proposed in 2004, “Internet gaming disorder” (IGD) was included in the *Diagnostic and Statistical Manual of Mental Disorders* (*DSM-5*) as a disorder requiring further research. IGD is prevalent in South Korea, and many studies have been conducted on the disorder. Previous studies have helped in understanding various aspects of IGD, but a comprehensive understanding of the research trends is required to identify research gaps. Therefore, we conducted a bibliometric review of all published IGD studies in South Korea. For the identification of articles, the Web of Science database was used. Data analysis was performed using Biblioshiny. A total of 330 publications were included in the analysis. The average number of citations per document was 17.12. These publications were written by a total of 658 authors, and the number of coauthors per document was 5.07. The years with the most publications were 2018 (*n* = 57), 2017 (*n* = 45), and 2019 (*n* = 40). The top three journals containing publications were the *Journal of Behavioral Addictions* (*n* = 46), *Frontiers in Psychiatry* (*n* = 19), and *Psychiatry Investigation* (*n* = 14). In a keyword analysis (apart from “IGD”, “internet addiction”, and “addiction”), the keywords “adolescent” (*n* = 31), “self-control” (*n* = 11), and “impulsivity” (*n* = 11) were included. T. This bibliometric analysis explores and summarizes the publications on IGD in South Korea. The results are expected to provide researchers with insights for further studies into IGD.

## 1. Introduction

While advances in web- and mobile-based technologies have enabled individuals to increase their productivity and improve social connection, the maladaptive and excessive use of technologies could in time result in psychosocial impairments among individuals of all age groups. The concept of Internet addiction was first proposed in 2004 by Kimberly Young [1]. Subsequently, as games have become more popular, the excessive playing of digital or video games has become more problematic. In 2013, the American Psychiatric Association’s *Diagnostic and Statistical Manual of Mental Disorders* (*DSM-5*) included Internet gaming as a condition that required further research. This is because there were insufficient data at that time to officially include it as a diagnosable condition [2]. The proposed criteria in the *DSM-5* state that for an individual to be considered to have the disorder, the extent of gaming must have had an impact on several domains of the individual’s life [2]. The proposed criteria included the following features: preoccupation, the presence of withdrawal symptoms when gaming is removed, intolerance and irritability when gaming is reduced or when the individual is attempting to quit, a narrowing of the repertoire of usual activities, having to lie to family members and friends about the extent of gaming, using gaming to alleviate negative moods, and relationship or employment difficulties due to gaming [2]. When Internet gaming was first considered a potential diagnostic criterion in the *DSM-5*, there were criticisms from experts in the field [3]. Kuss et al. [3], in their landmark paper, highlighted the numerous issues with the *DSM-5* conceptualization of gaming disorder and emphasized the importance for future research to carefully delineate between individuals who game excessively but without any psychosocial implications and individuals who game excessively to the extent that their functioning is affected. Since then, the International Classification of Diseases (ICD-11) has officially included gaming disorder in the diagnostic classification system [4]. To diagnose an individual with Internet gaming disorder (using the ICD-11), there has to be: (a) impaired control over gaming, (b) the prioritization of gaming over other activities, and (c) gaming despite there being negative consequences for the individual [4]. Some of the differences between the ICD-11 diagnostic criteria and the previous *DSM-5* diagnostic criteria include that they stipulate only a few essential criteria [5]. The ICD-11 criteria also removed the concepts of tolerance and withdrawal [5]. In the ICD-11, there is an emphasis on the need for functional impairments before a diagnosis can be made, which addresses the previous issues regarding differentiating between excessive users with and without resulting impairments in their life [5].

In a country such as South Korea, the Internet penetration rates are as high as 93% [6]. It is estimated that 47.32 million South Korean individuals make use of the Internet [6]. The gaming market in South Korea has been reported to be one of the largest in the world [7], with approximately 71% of the population having played video games in 2021 [8]. The estimated number of gamers in South Korea is over 35 million [8]. To date, there have been several studies examining the prevalence rates of Internet gaming disorder in South Korea, although there have been issues pertaining to the inclusion of gaming disorder in various diagnostic systems over the years. Yu et al. [9] previously sampled a total of 2024 students, and their gaming symptoms were further assessed against the *DSM-5* diagnostic criteria. They reported that 5.9% of the sample had gaming disorders, with males having a higher prevalence compared with females (10.4% versus 9.4%) [9]. Another 8% of those sampled were also deemed to be at high risk of gaming disorder [9]. In a recent meta-analysis that examined the prevalence rates of Internet gaming disorder in East Asia, the authors reported the overall pooled prevalence rates to be 12%, with a higher prevalence among males than females [10]. A close examination of the subgroup analysis within the meta-analysis showed that the pooled prevalence rate for South Korea was 11% (with nine studies contributing) [10]. The onset of the COVID-19 pandemic has also resulted in an increased reliance on digital technologies. Kim et al. [11] examined the impact of the pandemic on the gaming behaviour of adolescents in South Korea. They reported that individuals who met the “addictive gamer profile” were more susceptible to increasing their gaming time. Other studies have examined other aspects of Internet gaming disorder, such as changes in cognitive functioning among individuals with gaming disorder [12] and assessment of the current evidence for pharmacological and psychosocial treatments [13]. 

These aforementioned studies have provided an overview of the epidemiology of Internet gaming disorder in South Korea, along with findings relating to other aspects of the disorder. While the insights provided by the reviews have helped in understanding the evidence base of the field, it is important to conduct a quantitative bibliometric analysis of all the research related to Internet gaming disorder conducted to date to identify the research trends and the gaps in the research. Until now, to the best of our knowledge, there has only been one bibliometric analysis, undertaken by Tran et al. [14] in 2020, that has examined the trends in research involving excessive device and Internet use. One of the limitations of this prior bibliometric analysis was that it had a broad search strategy and included articles that examined topics from electronic devices to Internet addiction. Moreover, it is evident from the literature review that Internet gaming disorder is a condition that has been well studied in Korea. There has been no bibliometric review reported to date examining publications arising from Korea pertaining to Internet gaming disorder. The conduct of this review is of importance for the following reasons. The prevalence of gaming disorder in South Korea, based on a prior meta-analytical review by Liao et al. [10], is reported to be one of the highest in the world, at 11%. One of the reasons for the high rates in Korea might be the country’s huge gaming industry and the normalization of gaming in the population [15]. Findings arising from prior reviews, such as that by Darvesh et al. [16], also highlighted that there have been numerous publications from Korea. Therefore, it is timely for us to conduct this work to better understand the research trends and directions regarding gaming disorder in Korea. As the Internet penetration and smartphone ownership rates have increased in other parts of the world, Internet gaming disorder may become more prevalent in the rest of the world, following the case of Korea and therefore justifying the need for our current work. 

## 2. Methods

The Web of Science (WOS) database, used for the identification of relevant articles [17], was selected as it has been deemed superior relative to with Scopus and MEDLINE/PubMed databases. The WOS database not only allows for the extraction of articles with the full information, including titles, author names, total citations, and total download counts, but also covers all the citations of scientific publications since 1900, including all the high-impact scientific journals [18,19]. The WOS database includes several world-class indices, that is, the Social Sciences Citations Index and the Science Citation Index Expanded. The main strengths pertaining to the use of the WOS database are that it allows for reference tracing and citation reporting, as well as the visualisation of the research outputs in a specific area [19]. It is important to note that the WOS database remained the only broad-scope bibliometric database for more than 40 years, until 2004, when Scopus was launched [17]. The main investigators sought the assistance of a library information specialist (from the Lee Kong Chian School of Medicine, Nanyang Technological University) to craft an appropriate search strategy and to refine the strategy to ensure that all relevant articles were captured. Two independent search strategies were formulated: TS = ((Internet or computer or online) and (game$ or gaming)) and TS = ((game$ or gaming) NEAR/3 (disorder$ or addict* or problem* or excess*)). The strategies were then combined using the Boolean operator “AND”, and a search was conducted on 28 October 2022. TS refers to a search based on the topic of interest. NEAR/3 was used to find records whereby the terms “game or gaming” and “disorders, addiction, problem or excessive” are within three words of each other.

By means of this strategy, a total of 3860 publications were retrieved from the WOS database. The identified citations were then limited by their country of origin to that of “South Korea”. The citations were then exported into a plain text format file and converted into an Excel spreadsheet using Biblioshiny (RStudio, Version 1.4). Two authors (M.Z. and S.Y.) screened the database to identify relevant publications. Publications were included if they described any features of (a) Internet gaming addiction or (b) video game addiction or overuse and (c) had bibliometric information that was in English. Publications were excluded if (a) they merely described a computer game, (b) did not report on gaming disorder or the overuse of games, or (c) sampled a cohort that did not include Korean participants. Any disagreements were resolved by means of a discussion between the two authors. Figure 1 provides an overview of the publication selection process.

Data analysis was performed using Biblioshiny (R Studio, Version 1.4) [20]. This software helped in the analysis and visualisation of the following information: (a) the main information of the dataset, (b) the annual scientific production, (c) the journals that published the most recent sources, (d) the most cited authors, (e) the most cited countries, (f) the most cited global documents, (g) the most utilized keywords, and lastly, (h) co-occurrence and co-citation networks.

## 3. Results

### 3.1. Analysis of Publications

A total of 330 publications were included in this review. These publications were published between 2002 and 2022. A total of 134 sources contributed to these 330 publications. The average citations per document was 17.12, and the average citations per year per document was 2.635. Of these publications, 251 were articles, 3 were corrections, 2 were editorial materials, 1 was a letter, 56 were meeting abstracts, 6 were proceeding papers, and 11 were reviews. These publications included a total of 658 author keywords. Of the 330 publications, 22 were single-author publications. The average number of documents per author was 0.635, and the number of authors per document was 1.58. The number of coauthors per document was 5.07. The collaboration index was 1.63. Figure 2 provides an overview of the annual scientific publications by year. Most of the publications were published in 2018 (*n* = 57), 2017 (*n* = 45), or 2019 (*n* = 40). Figure 3 provides an overview of the three-field plot of authors, their affiliations, and the keywords used. This figure allows for the visualisation of the most productive authors, their affiliations, and the keywords they used. Only one of the authors is affiliated with an overseas university, i.e., the University of Utah. Apart from the keywords relating to Internet gaming disorder and Internet addiction, other leading keywords include impulsivity, adolescent, and self-control.

### 3.2. Journal Analysis

Table 1 provides further information regarding the top 20 most productive journals and the corresponding number of articles that were published in each of the journals. As is evident from the table, the top three journals in which most of the publications were published were the *Journal of Behavioral Addictions* (*n* = 46), *Frontiers in Psychiatry* (*n* = 19), and *Psychiatry Investigation* (*n* = 14). It is also clear that within the list, there are subspeciality journals and journals that are more general in nature. Table 2 provides an overview of the most productive authors in Korea (top 20). The authors Choi JS, Kim DJ, and Han DH contributed a total of 66, 66, and 52 articles, respectively.

### 3.3. Analysis of Citations

Table 3 provides an overview of the top 20 most cited publications, along with their corresponding matrix of total citations and citations per year. This table is of importance, as it helps to identify the most influential articles published by Koreans and their collaborators.

### 3.4. Analysis of Keywords

Figure 4 provides an overview of a word cloud based on the analysis of the authors’ keywords. This word cloud was generated with the number of words restricted to 50 and without any removal of keyword terminologies. Keyword analysis is helpful in identifying keyword trends. Apart from Internet gaming disorder, addiction, and Internet addiction, with frequencies of 111, 25, and 18, respectively, other keywords include adolescent (*n* = 16), adolescents (*n* = 15), self-control (*n* = 11), and impulsivity (*n* = 11). Figure 4 provides an overview of the co-occurrence of the authors’ keywords. A total of six different clusters can be identified. The keywords in cluster 1 include addiction, adolescent, Internet, game, smartphone, and stress; those in cluster 2 incorporate Internet addiction, game addiction, online gaming, and comorbidity; cluster 3 contains the keywords Internet gaming disorder, impulsivity, depression, functional connectivity, default mode network, behavioural addiction, FMRI (Functional Magnetic Resonance Imaging), heart rate variability, alcohol use disorder, game genre, grey matter volume, ADHD (Attention Deficit Hyperactivity Disorder), anterior cingulate cortex, anxiety, and children coherence; cluster 4 contains self-control, online game, aggression, bupropion, FMRI, dorsolateral prefrontal cortex, craving, major depressive disorder, and position emission tomography; cluster 5 includes online game addiction and self-esteem; and the cluster 6 keywords are adolescents, gaming disorder, and cohort. Figure 5 provides an overview of the co-occurrence of authors’ keywords. 

### 3.5. Analysis of Topic Trends

Figure 6 provides an overview of the thematic analysis of our topic of interest. A thematic map allows four typologies of themes to be defined based on the quadrant in which they are placed. Themes in the upper-right quadrant are known as motor themes, which are characterized by both high density and centrality, meaning that they are developed and relevant for the research field. Themes in the upper-left quadrant are niche themes, which are developed (high density) but isolated (low centrality). Themes in the lower-left quadrant are emerging or declining themes, as they have both low centrality and density, indicating that they are weakly developed and marginal. Themes in the lower-right quadrant are basic themes characterized by high centrality (relevant) and low density (less developed). Figure 7 provides an overview of the evolution of themes across the years. Slices were selected at three times: 2013, 2017, and 2020 (2013 corresponding to the *DSM-5* diagnosis, 2017-2019 to the discussion and formalisation of the ICD-11 diagnosis, and 2020 to the onset of COVID-19). This figure provides an overview of the changing trends in research through the years as the field develops.

## 4. Discussion

To the best of our knowledge, this is the first bibliometric review to have examined the publications arising from Korea regarding Internet gaming disorder. The conduct of this review is important for the reasons we have already highlighted in the Introduction, namely the high prevalence rates of gaming disorder and the number of publications in Korea pertaining to gaming disorder. Some of the key findings arising from this review include the fact that there was a substantial number of articles published from 2017 through to 2019. The results also suggest that authors have published in both subspecialized journals and general psychiatry journals. We postulate that the increase in the number of papers published after 2017 to 2019 might be due to the discussion and later on official introduction of gaming disorder as a diagnosable condition in the International Classification of Diseases (ICD-11) in 2019. Moreover, these time points coincided with the launch of the first cohort study examining Internet gaming disorder [41]. Jeong et al. [41], in their previously published study protocol, reported how the cohort study was first started in 2015 and the intention at the time was to recruit a total of 3000 adolescents. The outcomes that were examined were those of the 2-year incidence and the remission and recurrence rates for gaming disorder [41]. Some of the results from the cohort study were published in 2020; these results provided insights into how risk and preventive factors can be used for the prediction of the risk levels of gaming disorders [42]. With regard to the analysis of journals, we found that academics tend to publish in both subspecialized addiction journals and mainstream journals. Notably, *Psychiatry Investigation*, which is a Korean journal published by the Korean Neuropsychiatric Association, is among the top three most published journals. Articles were mostly published in the *Journal of Behavioral Addictions*, as it is one of the top journals in the field. The fact that other mainstream journals accept publications related to Internet gaming is encouraging, as behavioural addiction is a relatively niche field, and few journals are focused exclusively on behavioural health.

Our identification of the leading articles in the field provides insights into the most influential articles. Upon reviewing the list of top 20 studies, we found there were influential articles on prevalence (Mak KK et al., 2014) [22]; the risk factors associated with gaming (Kim EJ et al., 2008; Kwon JH et al., 2011; Choi SW et al., 2014; Koo HJ et al., 2014; Seo M et al., 2009) [21,26,29,31,40]; pharmacological treatments, such as investigations into the use of bupropion (Han DH et al., 2010; Han DH et al., 2012) [24,39] and methylphenidate (Han DH et al., 2009) [25]; and examinations of psychological therapies, such as cognitive behavioural therapy (Kim SW et al., 2012) [33] and family therapy (Han DH et al., 2012) [34]. Other influential articles were focused on exploring comorbid conditions such as depression (Wang HR et al., 2018) [37] and neuroimaging investigations (Han DH et al., 2010) [38]. This list of the top 20 articles highlights the different areas in the field of Internet gaming disorder that have been well studied and are currently influential in Korea. While Internet gaming disorder is a behavioural addiction, it is interesting to note that researchers have actively considered the use of pharmacological medications to examine whether there are any improvements in the clinical symptoms.

The word cloud and thematic analysis provide an overview of the developments in the field. Some of the fundamental and basic themes include investigation of impulsivity and functional connectivity and the effects of bupropion. Since the publication of initial studies by Han DH et al. [24] regarding the use of bupropion, Seo et al. [13] conducted a literature review examining the efficacy and related neural effects of both medications and therapeutic approaches for individuals with gaming disorder. They suggested that both bupropion and cognitive behavioural therapy could affect the resting-state functional connectivity within the cortico-subcortical circuit and the default mode network. It appears that in Korea, several niche areas have included explorations of the association between IGD and ADHD, the use of virtual reality for IGD, and the use of the auditory oddball task in determining whether there are impairments in auditory informational processing and cognitive capabilities. Regarding ADHD and IGD, a recent 3-year longitudinal study conducted by Lee et al. [43] demonstrated that those with comorbid ADHD have lower rates of recovery and higher levels of symptoms over time than those without ADHD. In a related study by Seo et al. [44], the authors reported how the rates of diagnosis of ADHD have changed since 2008 compared with 2018. In 2008, the prevalence rate for diagnosed ADHD was 127.1/100,000, and this increased to 192.9/100,000 by 2018. This might account for the existence of more studies examining the association between ADHD and other psychiatric disorders, such as gaming. Virtual reality has been investigated in Korea as a therapeutic intervention. Shin et al. [45] reported the development of a virtual Internet cafe and how this virtual cafe induced greater cravings in individuals with IGD compared with the control group. The authors therefore proposed that such a method could be used as part of cue exposure intervention. A more recent investigation was conducted by Shin et al. [46] in which they reported that a VR (virtual reality)-based application could help in managing game-related conflicts and in encouraging more desirable gaming behaviours. Consideration of the use of virtual reality tools as interventional tools may have arisen because there has been growing interest in the development of digital therapeutics in Korea, whereby software programs are now being considered as medical devices for treatment [47]. Regarding the auditory oddball task and IGD, prior studies, such as that by Choi, reported a negative correlation between the severity of IGD and the P300 EEG amplitudes and how a reduction in the P300 in an auditory oddball task might signify the presence of deficits in auditory information processing and cognitive capabilities [48]. From the overall thematic analysis, it is evident that research into IGD in Korea has been very diverse, ranging from investigations into the risk factors for IGD to neurobiological basis and even to the application of innovative technologies such as virtual reality to assist in treatment.

The findings from the thematic evolution map largely replicate the findings derived from the overall thematic map, with the exception that it helped to highlight that the main focus of most of research has been on adolescents. As far as we know, there is limited exploration of the existence of gaming disorder in other populations, such as in young adults or the elderly. Kim et al. [49], in their recently published paper, reported the impact of middle-aged parents’ Internet addiction on their children’s Internet gaming addiction. The authors reported that a child’s propensity to be addicted to the Internet increases if the parents’ level of Internet addiction is high [49]. Other studies, such as that by Jun et al. [50], have reported that smartphone addiction exists among the adult population in Korea. The rates of smartphone addiction in adults and senior citizens, based on the data presented by Jun et al. [50] for 2018, were 18.1% and 14.2%, respectively, compared with 29.3% in teenagers. It is possible that smartphone addiction is more of a concern for older adults, as per the findings of Jun et al. [50]. However, recent research conducted outside Korea revealed that IGD remains a problem among young adults. Zhang et al. [51] reported that excessive gaming among young adults might result in increased mind wandering and a non-productive state of mind. This demonstrates that the research conducted in Korea is limited at present, as studies have been focused on adolescent samples and have not examined other age groups.

As with other bibliometric analyses, the current study is subject to several limitations. While the usage of the WOS database has been justified previously, the reliance on this database alone may have led to the exclusion of some studies. We did not include or search through Korean journals, as we intended to examine only the Korean studies that are available internationally. We managed searched the database at only one time point, and more recent studies may have been published since then. As it is nowadays common for researchers to collaborate internationally, we were unable to remove articles that involved international collaborators, but we did remove any articles that did not include or examine a Korean sample. Furthermore, when the ICD-11 first formally included gaming as a diagnosable condition, it was met with some criticism within Korea, as in Korea, video gaming is a huge business, and according to previous reports, contributed more than USD 11.2 billion to the economy of South Korea in 2018 [52]. Our analysis of the publication trends did not specifically elucidate the impact of having a diagnosable condition on the gaming industry.

## 5. Conclusions

This bibliometric analysis explored IGD research trends in South Korea. From this research, overall publication trends including annual scientific production, authorship and co-authorship, citations, productive journals, productive authors, the top cited publications, keywords, topic trends, and the evolution of research themes were identified. The results are expected to provide researchers with insights into the gaps in the research and ideas for further studies on IGD. Additional research on IGD will provide more evidence for diagnosis and intervention.

## Figures and Tables

**Figure 1 ijerph-20-03786-f001:**
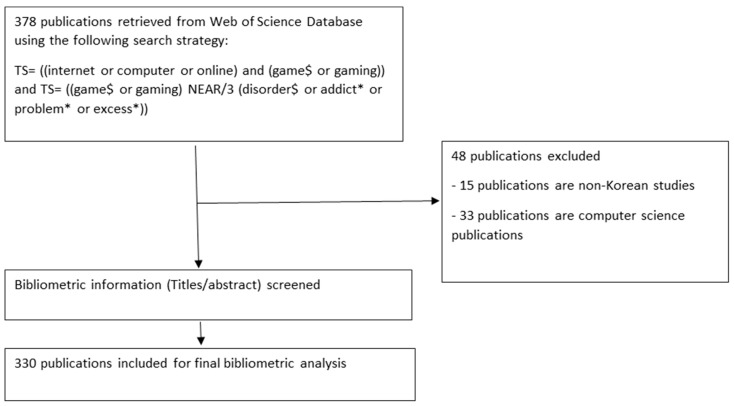
Overview of the study selection process.

**Figure 2 ijerph-20-03786-f002:**
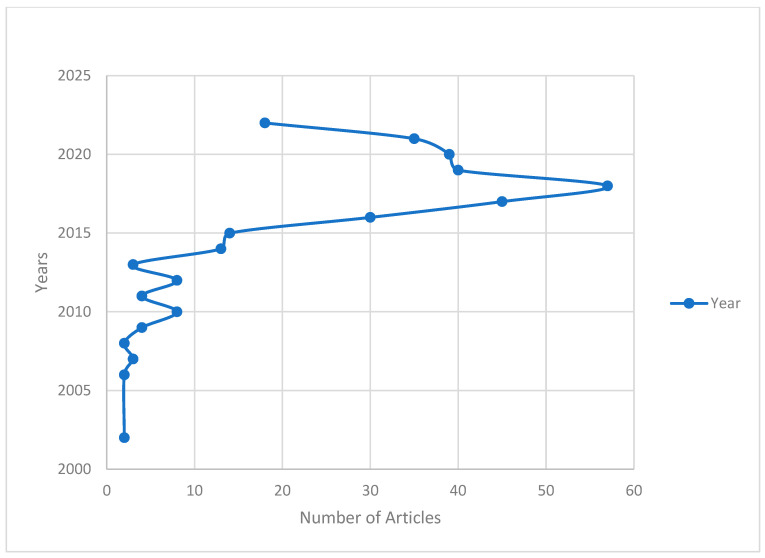
Annual scientific production.

**Figure 3 ijerph-20-03786-f003:**
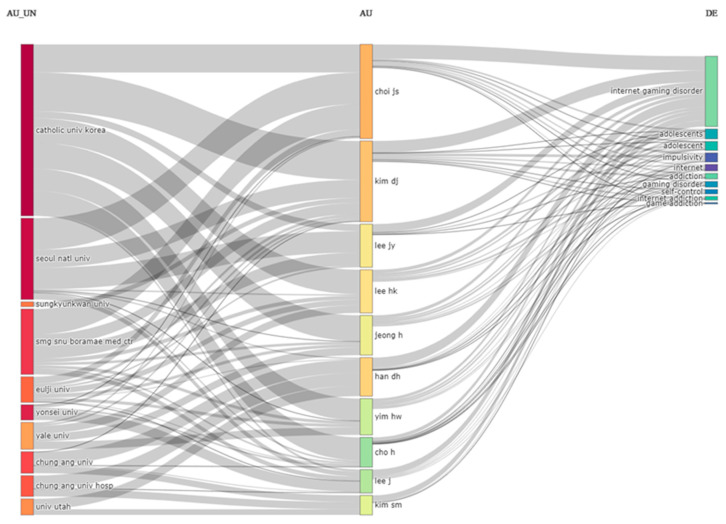
Three-field plot of authors, their affiliations, and keywords. Footnote: Each of the categories is limited to 10 items. The resolution of the image is limited by the software that is used for the processing of the image.

**Figure 4 ijerph-20-03786-f004:**
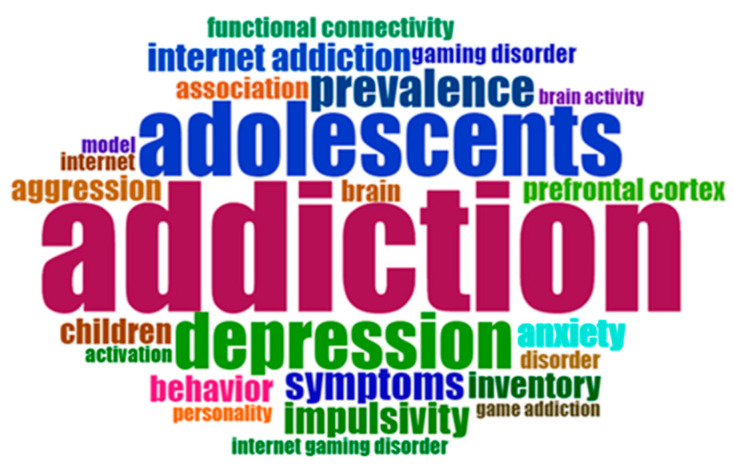
Word cloud of authors’ keywords.

**Figure 5 ijerph-20-03786-f005:**
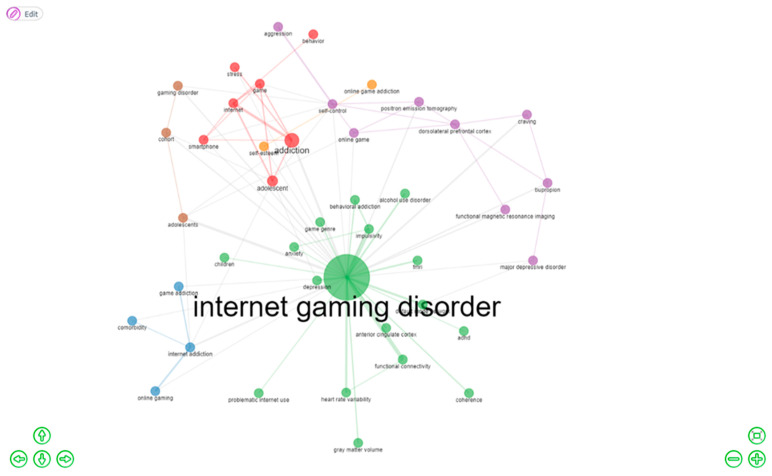
Co-occurrence network of authors’ keywords.

**Figure 6 ijerph-20-03786-f006:**
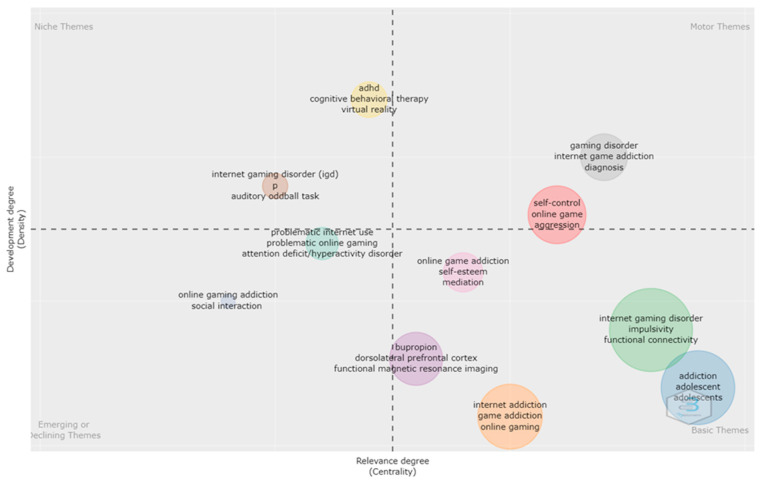
Thematic map.

**Figure 7 ijerph-20-03786-f007:**
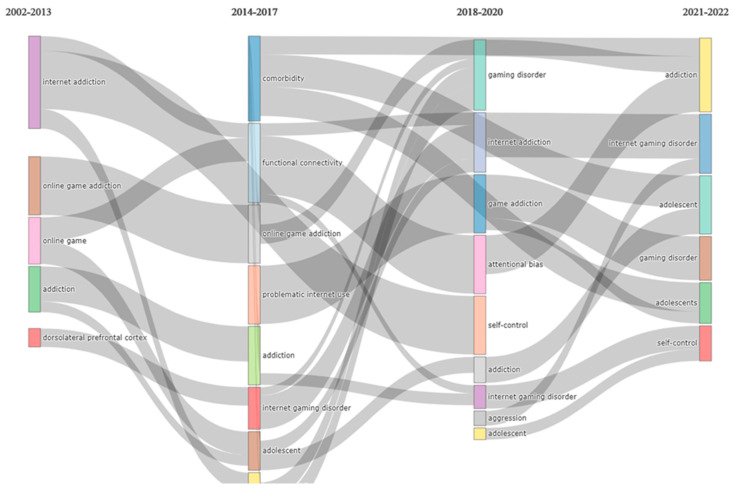
Overview of the evolution of themes across the years.

**Table 1 ijerph-20-03786-t001:** The top 20 most productive journals.

Journal	Articles
*JOURNAL OF BEHAVIORAL ADDICTIONS*	46
*FRONTIERS IN PSYCHIATRY*	19
*PSYCHIATRY INVESTIGATION*	14
*CYBERPSYCHOLOGY BEHAVIOR AND SOCIAL NETWORKING*	12
*EUROPEAN PSYCHIATRY*	10
*SCIENTIFIC REPORTS*	9
*ADDICTION BIOLOGY*	8
*COMPUTERS IN HUMAN BEHAVIOR*	7
*INTERNATIONAL JOURNAL OF ENVIRONMENTAL RESEARCH AND PUBLIC HEALTH*	7
*ALCOHOLISM: CLINICAL AND EXPERIMENTAL RESEARCH*	6
*JOURNAL OF CLINICAL MEDICINE*	6
*ALCOHOL AND ALCOHOLISM*	5
*PROGRESS IN NEURO-PSYCHOPHARMACOLOGY & BIOLOGICAL PSYCHIATRY*	5
*ADDICTIVE BEHAVIORS*	4
*INTERNATIONAL JOURNAL OF NEUROPSYCHOPHARMACOLOGY*	4
*JOURNAL OF KOREAN MEDICAL SCIENCE*	4
*PLOS ONE*	4
*TRANSLATIONAL PSYCHIATRY*	4
*CHILDREN AND YOUTH SERVICES REVIEW*	3
*FRONTIERS IN PSYCHOLOGY*	3

**Table 2 ijerph-20-03786-t002:** The top 20 most productive authors in Korea.

Author	Articles
CHOI JS	66
KIM DJ	66
HAN DH	52
LEE HK	32
LEE JY	30
JEONG H	28
KIM SM	27
YIM HW	27
LEE J	26
CHO H	25
LEE SY	23
LEE D	22
PARK M	22
JO SJ	21
JUNG YC	20
KIM H	19
RENSHAW PF	19
CHUN JW	18
LEE YS	18
KWEON YS	16

**Table 3 ijerph-20-03786-t003:** Overview of the top 20 most cited publications.

Author, Year of Publication, Journal	Title of Article	Total Citations	Total Number of Citations per Year
KIM EJ, 2008, *EUR PSYCHIAT* [21]	The relationship between online game addiction and aggression, self-control, and narcissistic personality traits	344	22.93
MAK KK, 2014, *CYBERPSYCH BEH SOC N* [22]	Epidemiology of Internet Behaviors and Addiction Among Adolescents in Six Asian Countries	226	25.11
JEONG SH, 2016, *COMPUT HUM BEHAV* [23]	What type of content are Smartphone users addicted to?: SNS vs. games	216	30.86
HAN DH, 2010, *EXP CLIN PSYCHOPHARM* [24]	Bupropion sustained release treatment decreases craving for video games and cue-induced brain activity in patients with Internet video game addiction	168	12.92
HAN DH, 2009, *COMPR PSYCHIAT* [25]	The effect of methylphenidate on Internet video game play in children with Attention-deficit/hyperactivity disorder	162	11.57
KWON JH, 2011, *COMMUNITY MENT HLT J* [26]	The Effects of Escape from Self and Interpersonal Relationship on the Pathological Use of Internet Games	127	10.58
HAN DH, 2007, *J ADDICT MED* [27]	Dopamine genes and reward dependence in adolescents with excessive Internet video game play	117	7.31
KIM NR, 2016, *PSYCHIAT INVEST* [28]	Characteristics and Psychiatric Symptoms of Internet Gaming Disorder among Adults Using Self-Reported DSM-5 Criteria	112	16.00
CHOI SW, 2014, *J BEHAV ADDICT* [29]	Similarities and differences among Internet gaming disorder, gambling disorder and alcohol use disorder: a focus on impulsivity and compulsivity	103	11.44
HYUN GJ, 2015, *COMPUT HUM BEHAV* [30]	Risk factors associated with online game addiction: a hierarchical model	98	12.25
KOO HJ, 2014, *YONSEI MED J* [31]	Risk and protective factors of Internet addiction: a meta-analysis of empirical studies in Korea	92	10.22
KIM MG, 2010, *COMPUT HUM BEHAV* [32]	Cross-validation of reliability, convergent and discriminant validity for the problematic online game use scale	92	7.08
KIM SM, 2012, *COMPUT HUM BEHAV* [33]	Combined cognitive behavioral therapy and bupropion for the treatment of problematic on-line game play in adolescents with major depressive disorder	88	8.00
HAN DH, 2012, *PSYCHIAT RES-NEUROIM* [34]	The effect of family therapy on the changes in the severity of on-line game play and brain activity in adolescents with on-line game addiction	87	7.91
LEE MS, 2007, *CYBERPSYCHOL BEHAV* [35]	Characteristics of Internet use in relation to game genre in Korean adolescents	81	5.06
JO YS, 2019, *J CLIN MED* [36]	Clinical Characteristics of Diagnosis for Internet Gaming Disorder: Comparison of DSM-5 IGD and ICD-11 GD Diagnosis	79	19.75
WANG HR, 2018, *J AFFECT DISORDERS* [37]	Prevalence and correlates of comorbid depression in a nonclinical online sample with DSM-5 Internet gaming disorder	75	15.00
HAN DH, 2010, *CYBERPSYCH BEH SOC N* [38]	Changes in Cue-Induced, Prefrontal Cortex Activity with Video-Game Play	73	5.62
HAN DH, 2012, *J PSYCHOPHARMACOL* [39]	Bupropion in the treatment of problematic online game play in patients with major depressive disorder	66	6.00
SEO M, 2009, *CIN-COMPUT INFORM NU* [40]	Internet addiction and Interpersonal problems in Korean adolescents	66	4.71

## Data Availability

No new data is created as all the citations were retrieved from Web of Science.

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
