# Peer review of "A Bibliometric Analysis of Research into Internet Gaming Disorders in Korea"

_ijerph, 2023, doi:10.3390/ijerph20053786_

Round 1

Reviewer 1 Report

This article is well described, structured and understandable, but I don't see any interest to publish it in an international journal.

The authors have to send it to a Korean journal, I don't see how this article could be of interest to the international scientific community despite its editorial quality.

Author Response

Dear Reviewer 1,

Thank you for reviewing our work, and for your positive comments. While the intent of this manuscript is essentially in identifying articles published by Korean authors and might appear to have limited relevance to an international journal, most of the published works by Korean authors are in international journals, such as the Journal of Behavioural Addictions and Frontiers in Psychiatry. There is thus a need to map out the existing Korean research and this allows further comparison with the research done in other countries/or by other collaborators. Also, Korea has a relatively high Internet penetration rates, and with a substantial population of Internet gamers. The insights provided by this article would help researchers in other countries plan and conceptualise their research work, learning from the experience of Korea, given the relatively high incidence of gaming issues in Korea. It should be noted that some of the articles might also be published in collaboration with other international collaborators. Given these reasons, we opined that this article remains of relevance to the international community and might help in fostering further research collaborators with Korean academics.

Reviewer 2 Report

The authors propose a bibliometric analysis on the phenomenon of Internet Gaming Disorder, limited to the diffusion of studies in South Korea. They only referred to the Web Of Science online database and the analysis was performed using Biblioshiny.

The research is sufficiently well conducted and it has mostly correclty followed the guide lines for bibliometric analyses, although systematic searches were limited to one online database.  As a whole, the study is sufficiently understandable, but English is definitely to be improved, particularly in sentence fluency.

Comments:

Lines 34-38: the logical reasoning of the sentence is not very clear. Either DSM 5 has included Gaming Disorder as a separate disorder with defined diagnostic criteria, or there is insufficient data to define it. I would rephrase the sentence, pointing to the DSM 5 diagnostic criteria, and only then emphasize the indicated suggestion to investigate the phenomenon further.

Lines 59-62: I would rephrase the sentence, since it is currently hardly understandable. Furthermore, I would be more cautious in describing the reduction of diagnostic criteria as " advantages.” (line 56): the number of diagnostic criteria is not in itself indicative of a more or less correct diagnosis or implementation.

Line 84: type “ta”. Overall, I’d rephrase the sentences with a more fluent English.

Line 106: I do not understand the repetition of the comparison with China and Hong Kong, unless there are reasons not made explicit in the text.

Lines 111-114: the sentence is not clear.

129-136, 137-140: this part is redundant and does not add useful information to the study.

140-144: the search string is not clearly stated. Furthermore, “NEAR/3” is not explained.

Figure 1: check if the Figure is correct, the search string is truncated. A good example can be retrieved from the updated PRISMA guidelines.

Table 4: I see no particular reason to include four decimal places in the “Number of Total Citations by year” column.

Line 307: typo “manging”.

Lines 327-330: this postulation requires adequate citation support.

Lines 342-343: if a considerable amount of time has passed between the first systematic search and the submission of the study to the journal, a second systematic search can be performed to find any new studies.

Author Response

Dear Reviewer 2,

Thank you for peer reviewing our work. This is correct. We have conducted a bibliometric review using Web of Science database, and the analysis was performed by using Biblioshiny package in R.

The research is sufficiently well conducted and it has mostly correclty followed the guide lines for bibliometric analyses, although systematic searches were limited to one online database.  As a whole, the study is sufficiently understandable, but English is definitely to be improved, particularly in sentence fluency.

Thank you for your positive comments. Please refer to our methods for our justification as to why we have selected Web of Science Database. We have chosen Web of Science as this database was deemed to be superior in comparison to Scopus or MEDLINE/PubMED. WOS allows for the extraction of comprehensive information regarding each of the publications, ranging from the titles, authors names, total citations, total download counts, and it has a wide coverage from 1900. Most importantly, the WOS database also allows for reference tracing and citation reporting. We have also added on a reference that provided further justifications that WOS has been a database that has been in existence prior to SCOPUS.

We apologise for the language issues, and in this revised manuscript, we have taken on board your recommendations and addressed them accordingly.

Comments:

Lines 34-38: the logical reasoning of the sentence is not very clear. Either DSM 5 has included Gaming Disorder as a separate disorder with defined diagnostic criteria, or there is insufficient data to define it. I would rephrase the sentence, pointing to the DSM 5 diagnostic criteria, and only then emphasize the indicated suggestion to investigate the phenomenon further.

We have amended the sentence to “In 2013, the American Psychiatric Association’s Diagnostic and Statistical Manual of Mental Disorders (DSM5) included Internet Gaming as a condition that required further research. This is because there was insufficient data then to officially include it as a diagnosable condition [2].”

Lines 59-62: I would rephrase the sentence, since it is currently hardly understandable. Furthermore, I would be more cautious in describing the reduction of diagnostic criteria as " advantages.” (line 56): the number of diagnostic criteria is not in itself indicative of a more or less correct diagnosis or implementation.

We have amended the sentences to “To diagnose an individual with Internet Gaming Disorder (by the ICD-11) criteria, one has to have (a) impaired control over gaming, (b) have prioritized gaming over other activities and (c) been gaming despite there being negative consequences for the individual [4]”. We have reworded and removed the use of the terminology “advantages”. Instead, we have stated that it differed from the DSM-5. The amended sentence is as follows: “Some of the differences that the ICD-11 diagnostic criteria have had over the previous DSM-5 diagnostic criteria was that it has stipulated only a few essential criterions.[5]. It has also removed the concepts of tolerance and withdrawal [5]. In the ICD-11, there has been emphasis on the need for there to be functional impairments before the diagnosis can be made, which would address the previous issues regarding differentiating excessive users with and without results impairments in their life [5].”

Line 84: type “ta”. Overall, I’d rephrase the sentences with a more fluent English.

We have corrected the typo mistake. The sentence has been reworded to “They reported that individuals who met the “addictive gamer profile” were more susceptible towards increasing their gaming time.”

Line 106: I do not understand the repetition of the comparison with China and Hong Kong, unless there are reasons not made explicit in the text.

We have removed the prevalence rates reported to avoid any confusion. We have included the citation to provide insights into the prevalence rates of gaming disorder in Korea. The amends are as follows: “A close examination of the sub-group analysis within the meta-analysis showed that the pooled prevalence rate for South Korea was that of 11% (with 9 studies contributing).”

Lines 111-114: the sentence is not clear.

We have amended the sentence to “One of the reasons for the high rates in Korea might be because of Korea’s huge gaming industry, and the normalization of gaming in the population. [15].”

129-136, 137-140: this part is redundant and does not add useful information to the study.

We have removed this part as per your suggestion.

140-144: the search string is not clearly stated. Furthermore, “NEAR/3” is not explained.

We seek to clarify that we have implemented the search string as listed out in the methods. We have provided an explanation of the NEAR/3 terminology. The amends are “ NEAR/3 was used to find records whereby the terms “game or gaming” and “disorders, addiction, problem or excessive) are within 3 words of each other.

Figure 1: check if the Figure is correct, the search string is truncated. A good example can be retrieved from the updated PRISMA guidelines.

We have adjusted Figure 1 such that the words are not truncated. In addition, we have also referenced PRISMA guidelines and added on more information into Figure 1.

Table 4: I see no particular reason to include four decimal places in the “Number of Total Citations by year” column.

We have amended the total citations by year figures to 2 decimal places.

Line 307: typo “manging”.

We have corrected this.

Lines 327-330: this postulation requires adequate citation support.

We have removed that postulation. Instead, we have highlighted a prior study sampling smartphone addiction in adults and discuss how smartphone addiction might be more of a concern for older adults, given the findings of the previous study. We then discussed how other countries have sampled Internet gaming in adults and argue that Korean researchers ought to consider doing so. The amends are "Other studies such as that of Jun et al. [52] have reported that smartphone addiction exists amongst the adult population in Korea. The rates of smartphone addiction for adults and senior citizens, based on the data presented by Jun et al. [52] for 2018, were that of 18.1% and 14.2% respectively, as compared to 29.3% in teenagers. It might be that smartphone addiction is more of a concern for older adults, as per Jun et al. [52]’s findings. However, there have been recent research done outside of Korea, that revealed that IGD remains a problem among young adults”.

Lines 342-343: if a considerable amount of time has passed between the first systematic search and the submission of the study to the journal, a second systematic search can be performed to find any new studies.

We have indeed stated that a cross-sectional search was a limitation of this bibliometric review, but to be honest, any systematic review and meta-analysis remains to be limited by the same constraint. In view of resource limitations, it is difficult for us to keep on re-performing the search and conducting re-analysis of the results, as the paper undergoes subsequent reviews. It is unlikely that new studies would have a dramatic impact on the presented results, given that this is a bibliometric analysis, and the main data used for analysis are mainly the citations indexes of each of the papers. New papers are unlikely to accumulate as many citations, given that they have been recently published, and hence would not affect the results significantly.

Reviewer 3 Report

The manuscript reports a bibliometric review analysis of studies regarding Internet gaming disorder in South Korean context. In general, the study is well conducted, and the methodological choices seem to be in line with general guidelines of bibliometric reviews. I have some remarks and suggestions that I think could improve the manuscript. In addition, the text needs a thorough language check by a native English speaker as there were multiple grammatical and typographical errors throughout the text. I also spotted some inconsistencies between numbers in the main text vs. tables. Please find below more detailed comments.

INTRODUCTION

  • Define what is meant by "gaming" in this context when talking about gaming disorder and IGD (e.g. in relation to "gambling").
  • ICD-11 -> International Classification of Diseases (not "Disease")
  • The introduction could be shortened and reorganized for clarity. For example, there is some repetition (e.g., regarding gaming prevalence rates). I also suggest ending the Introduction by defining the aims/goals/research questions of this study before heading to the methods sections.

  METHODS

  • Page 3, line 28: Please provide references/published evidence when making this statement: "The WOS database was selected, as it has been deemed to be superior in comparison to that of Scopus or MEDLINE/PubMed."
  • TS and NEAR/3 -> please define more clearly what these commands stand for.
  • Figure 1: I think there are some missing text in the upper and in the rightmost boxes.
  • I assume that all the included studies were written in English but please specify whether this was an inclusion criterion.

  RESULTS

  • Table 1: consider using a figure (e.g. a line graph) instead of a table when showing annual production of articles.
  • 3.2. Journal analysis: the number of articles published in the journals Journal of Behavioral Addictions and  Psychiatry Investigation do not match in the main text and Table 2. Please double check these numbers (also in the abstract).
  • Page 7 line 199 "...and citations per day" -> I believe this should say "citations per year" as in Table 4.

  DISCUSSION

  • Even though you have elaborated the reasons for conducting this study already in the introduction, I would still shortly provide the main reasons and contributions also in the discussion.
  • Page 11 line 263: The journal name (Psychiatry Investigation) should be capitalized.

  OTHER:

  • The manuscript needs a proper language check as well as double checking all the numbers and their correspondence appearing in the abstract, main text, and tables.
  • In the main text, write journal names in italics (cursive).
  • In the reference list, consider marking the articles that were included in the data differently from other reference articles (for example with an asterisk*)

Author Response

Thank you, Reviewer 3, for peer reviewing our work. Thank you for your positive comments. We have taken your recommendations into consideration and revised this manuscript substantially. Please find our in-line replies to your comments.

INTRODUCTION

  • Define what is meant by "gaming" in this context when talking about gaming disorder and IGD (e.g. in relation to "gambling").

We have included a sentence to state that Gaming could refer to either video or digital games. We do not feel that it is appropriate to include any references to gambling, as this is not the topic of interest that we wish to focus upon in this work and might confuse our readers. The amends are as follows: “Subsequently, as games became more popular, the excessive playing of digital or video games could be problematic.”

  • ICD-11 -> International Classification of Diseases (not "Disease")

We have amended this and changed Disease to Diseases.

  • The introduction could be shortened and reorganized for clarity. For example, there is some repetition (e.g., regarding gaming prevalence rates). I also suggest ending the Introduction by defining the aims/goals/research questions of this study before heading to the methods sections.

Thank you for your suggestion. In the second paragraph of the introduction, we have removed the prevalence rates of gaming disorder in other countries and focused on reporting only the rates in Korea. We have modified the last paragraph of the introduction, such that it ends off with the main aim of this study. We have removed this repeated information/justification, “As such, it is important for the conduct of a bibliometric review, as such a review allows for the identification of research trends in Korea relating to gaming disorders and is helpful as well in identifying potential gaps in the field. Bibliometric review has been deemed to be a useful and rigorous method, that allow scientists to explore large dataset, and allows for the extraction of quantitative information by author, time, country, and journal [19].”

  METHODS

  • Page 3, line 28: Please provide references/published evidence when making this statement: "The WOS database was selected, as it has been deemed to be superior in comparison to that of Scopus or MEDLINE/PubMed."

We have now included the right reference to justify this statement. The reference is “PranckutÄ—, R. Web of Science (WoS) and Scopus: The Titans of Bibliographic Information in Today’s Academic World. Publications 20219, 12. https://doi.org/10.3390/publications9010012”. We have stated the rationale as to why WOS is deemed to be slightly superior as compared to SCOPUS. The amends are “It remains important to note that WOS remains to be the first broad scope bibliometric database for more than 40 years, until 2004, where Scopus was then launched [19].”

  • TS and NEAR/3 -> please define more clearly what these commands stand for.

We have now included definitions for both TS and NEAR/3. They are as follows “TS refers to a search based on the topic of interest. NEAR/3 was used to find records whereby the terms “game or gaming” and “disorders, addiction, problem or excessive) are within 3 words of each other.”

  • Figure 1: I think there are some missing text in the upper and in the rightmost boxes.

We have now modified Figure 1 and fixed the issues relating to the truncated text.

  • I assume that all the included studies were written in English but please specify whether this was an inclusion criterion.

We have included a specifier that the included studies were required to have bibliometric information that were in English language.

  RESULTS

  • Table 1: consider using a figure (e.g. a line graph) instead of a table when showing annual production of articles.

We have removed Table 1 and included a Figure (Figure 2) instead. This figure shows the number of articles published per year.

  • 3.2. Journal analysis: the number of articles published in the journals Journal of Behavioral Addictions and  Psychiatry Investigation do not match in the main text and Table 2. Please double check these numbers (also in the abstract).

We have double checked the data and the data within the table is correct. We have amended the figures reported in the main text accordingly.

  • Page 7 line 199 "...and citations per day" -> I believe this should say "citations per year" as in Table 4.

We apologized for the typo and have revised this to citations per year.

  DISCUSSION

  • Even though you have elaborated the reasons for conducting this study already in the introduction, I would still shortly provide the main reasons and contributions also in the discussion.

We have included this. The amends are “The conduct of this review is of importance for the reasons we have already highlighted in the introduction, which are namely the high prevalence rates of gaming disorder and the number of publications in Korea pertaining to gaming disorder.”

  • Page 11 line 263: The journal name (Psychiatry Investigation) should be capitalized.

We have amended this.

  OTHER:

  • The manuscript needs a proper language check as well as double checking all the numbers and their correspondence appearing in the abstract, main text, and tables.

We have submitted the revised manuscript for MDPI English editing service.

  • In the main text, write journal names in italics (cursive).

We have done so.

  • In the reference list, consider marking the articles that were included in the data differently from other reference articles (for example with an asterisk*)

We have done so.

Round 2

Reviewer 1 Report

thanks to the authors for their additions and comments

Author Response

We thank you Reviewer 1 for your continued peer review of our work. We're glad that we have addressed your comments/concerns. 

Reviewer 2 Report

I thank the Authors for the corrections made to the study. I believe that, as it stands, it is sufficiently adequate for publication. It is a methodologically sound study and makes important contributions to scientific research, both in the Korean and international fields.

Author Response

Thank you Reviewer 2 for your kind comments, and for recognizing the quality of our work.